# Characterization of MET Alterations in 37 Gastroesophageal Cancer Cell Lines for MET-Targeted Therapy

**DOI:** 10.3390/ijms25115975

**Published:** 2024-05-29

**Authors:** Jin-Soo Kim, Mi Young Kim, Sungyoul Hong

**Affiliations:** 1Department of Internal Medicine, Seoul National University Boramae Medical Center, Seoul 07061, Republic of Korea; moung9805@naver.com; 2College of Pharmacy, Seoul National University, Seoul 08826, Republic of Korea; sungyoul@snu.ac.kr

**Keywords:** gastric cancer, proto-oncogene protein c-met, tyrosine kinase inhibitors, cell line, tumor

## Abstract

Capmatinib and savolitinib, selective *MET* inhibitors, are widely used to treat various *MET*-positive cancers. In this study, we aimed to determine the effects of these inhibitors on *MET*-amplified gastric cancer (GC) cells. Methods: After screening 37 GC cell lines, the following cell lines were found to be *MET*-positive with copy number variation >10: SNU-620, ESO51, MKN-45, SNU-5, and OE33 cell lines. Next, we assessed the cytotoxic response of these cell lines to capmatinib or savolitinib alone using cell counting kit-8 and clonogenic cell survival assays. Western blotting was performed to assess the effects of capmatinib and savolitinib on the *MET* signaling pathway. Xenograft studies were performed to evaluate the in vivo therapeutic efficacy of savolitinib in MKN-45 cells. Savolitinib and capmatinib exerted anti-proliferative effects on MET-amplified GC cell lines in a dose-dependent manner. Savolitinib inhibited the phosphorylation of *MET* and downstream signaling pathways, such as the protein kinase B (AKT) and extracellular signal-regulated kinase (ERK) pathways, in *MET*-amplified GC cells. Additionally, savolitinib significantly decreased the number of colonies formed on the soft agar and exerted dose-dependent anti-tumor effects in an MKN-45 GC cell xenograft model. Furthermore, a combination of trastuzumab and capmatinib exhibited enhanced inhibition of AKT and ERK activation in human epidermal growth factor receptor-2 (*HER2*)- and *MET*-positive OE33 cells. Targeting MET with savolitinib and capmatinib efficiently suppressed the growth of *MET*-amplified GC cells. Moreover, these MET inhibitors exerted synergistic effects with trastuzumab on *HER2*- and *MET*-amplified GC cells.

## 1. Introduction

Gastric cancer (GC) is among the most common cancers worldwide, with more than one million new cases reported annually [1] and figures suggesting it is the 3rd most deadly cancer, with an estimated 783,000 deaths in 2018 [2]. During the last few decades, the efficacy of chemotherapy for GC has only advanced modestly, and targeted therapies have shown few successes. Many human epidermal growth factor receptor-2 (*HER2*)-targeting agents, such as lapatinib, pertuzumab, and T-DM1, have been tested; however, to date, only trastuzumab (Tmab) has been approved for *HER2*-positive GC treatment [3]. To dissect the genomic pathophysiology of GC in greater depth, The Cancer Genome Atlas (TCGA) reported GC subgroups and has shown that the disease can be categorized into four genomic subtypes and suggested several candidates for targeted therapy [4]. For example, a new agent targeting claudin-18 isoform 2 (CLDN18.2) showed promise for *HER2*-negative, CLDN18.2 + GC patients in the recent SPOTLIGHT trial [5]. The monoclonal antibody zolbetuximab significantly prolonged progression-free survival (PFS) and overall survival (OS) when combined with mFOLFOX6 compared to placebo plus mFOLFOX6.

Other potential targets in GC include the hepatocyte growth factor (HGF)–hepatocyte growth factor receptor (c-MET) pathway. The *MET* gene encodes the hepatocyte growth factor receptor (*HGFR*), also known as the *MET* receptor, which is a receptor tyrosine kinase involved in cell growth, survival, and migration [6]. Analysis of 444 patients with GC showed that the positivity determined by immunohistochemistry (IHC) for c-MET was 24.8% and the *MET* amplification rate was 2.3% [7]. Patients with GC and *MET* amplification may have a worse prognosis than those without these genetic alterations [8]. However, the criteria for defining MET protein overexpression have varied across numerous studies. In contrast to initial phase II studies, phase III trials did not demonstrate any clinical benefit from anti-MET monoclonal antibodies (mAbs) in GCs, even in patients with MET-positive disease in the RILOMET-1 [9] and METGastric [10] trials. The primary limitation of these trials was the inclusion of patients in whom the MET did not clearly drive the disease. As a result, the trials with the highest expression of *MET* gene amplification had fewer participants, potentially explaining the observed negative outcomes [11].

There are many *MET*-targeted agents in preclinical and clinical development. Using a molecular hybridization approach combined with a macrocyclization strategy for structural optimization, researchers developed D6808 [12]. This compound exhibited promising activity specifically targeting *MET*-amplified Hs746T gastric cancer cells in vitro. Others reported KRC-00715 as a selective inhibitor of *MET* which showed excellent efficacy in Hs746T GC cells in vitro and in vivo [13]. ABN401 is a potent and highly selective *MET* inhibitor and shows a favorable PK profile [14]. Currently, clinical trials are under way involving ABN401 for various types of cancers with *MET* alterations. SAR125844 has shown promise in preclinical studies for the treatment of *MET*-amplified GC [15], and a phase I study including GC patients was completed [16]. Unfortunately, clinical developments with SAR125844 were not active. Tivantinib, originally reported as a MET TKI, has been proposed as a potential inhibitor of VEGF signaling and MYC expression in GC cells that express VEGF-A, such as SNU620 and MKN45 [17]. Tivantinib as a monotherapy showed a modest efficacy in previously treated GC patients [18]. However, subsequent trials in various types of cancers failed to demonstrate sufficient efficacy and safety, leading to the termination of its clinical development.

Since the failure of these mAbs against MET, many tyrosine kinase inhibitors (TKIs) against *MET* have been developed for various types of cancers, including GC. Several studies have consistently used FISH to define *MET* amplification, and recent trials of MET inhibitors for *MET*-amplified cancers have been successful. As previously mentioned, numerous attempts to develop targeted agents for *MET* in GC have proven unsuccessful. This could originate from either the lack of efficacy or safety of the investigational agents, or from challenges in defining the appropriate biomarker for MET-targeted agents. For non-small cell lung cancers (NSCLCs), capmatinib [19], tepotinib [20], and crizotinib [21] are currently recommended for patients with high-level *MET* amplification or *MET* exon 14 skipping mutations in the US. Savolitinib was conditionally approved for the treatment of NSCLC with *MET* exon 14 skipping alterations in China. These approved *MET* TKIs have confirmed efficacy and could also be effective in patients with GC, if we can define the right biomarker for *MET* alterations. 

This study aimed to evaluate the efficacy of capmatinib and savolitinib against *MET*-amplified GC cell lines in vitro and to test savolitinib in a xenograft model.

## 2. Results

### 2.1. Identification of MET-Amplified GC Cells and Assessment of Their Sensitivity to Capmatinib and Savolitinib

We evaluated 37 GC cell lines for *MET* amplification using droplet digital polymerase chain reaction (ddPCR). MET protein overexpression was not significantly correlated with *MET* amplification in these cells (Appendix A). MET overexpression is not the right biomarker to identify *MET*-positive GC cells for *MET* TKIs in clinical trials [11]. Here, ddPCR revealed that five GC cell lines exhibited *MET* copy numbers >10 (Figure 1a). Subsequently, CCK-8 assays were performed on cells treated with capmatinib (Figure 1b) and savolitinib (Figure 1c). *MET*-amplified GC cells exhibited dose-dependent responses to these agents. Notably, NUGC-4 and SNU-638 cells, which exhibited MET overexpression but not amplification, only showed modest responses to *MET* TKIs. 

### 2.2. Clonogenic Cell Survival Assay Revealed That Capmatinib and Savolitinib Inhibited MET-Positive GC Cell Growth

Next, we performed clonogenic cell survival assays on MKN45, OE33, SNU638, and AGS cells. After 2 weeks of culture, MET-positive GC (MKN45 and OE33) cells showed a significant decrease in colony formation in a dose-dependent manner after both capmatinib and savolitinib treatments (Figure 2a,b). AGS cells (human gastric adenocarcinoma-derived cells) were selected as the *MET*-negative models. Notably, MET-overexpressing GC (SNU-638) cells, but not AGS cells, were responsive to *MET* TKIs. Although other *MET*-positive GC (SNU-5, SNU-620, and ESO51) cells showed good responses to *MET* TKIs, MET-overexpressing GC (NUGC-4) cells were only modestly sensitive to *MET* TKIs (Appendix A). Overall, MET-overexpressing GC cells showed variable sensitivity, whereas *MET*-amplified GC cells showed high sensitivity to *MET* TKIs.

### 2.3. Savolitinib Inhibits the Phosphorylation of MET and Its Downstream Molecules, Protein Kinase B (AKT), Extracellular Signal-Regulated Kinase (ERK), and S6

Savolitinib exhibited promising activity in Chinese patients with advanced NSCLC with *MET* ex14 skipping alterations, including pulmonary sarcomatoid carcinoma, in a phase II trial [22]. Based on the results of this trial, on 22 June 2021, savolitinib was conditionally approved for the treatment of NSCLC with *MET* ex14 skipping alterations in patients who have progressed or are intolerant to platinum-based chemotherapy in China. The anti-proliferative effects of the drug are mediated by the inhibition of phosphorylated c-Met and downstream signaling via the ERK and AKT pathways [23]. To evaluate the effect of savolitinib on *MET*-amplified GC cells, the signals related to MET expression were assessed via Western blotting. We treated these GC cells with 1, 10, and 100 nM of savolitinib for 24 h. Savolitinib effectively inhibited MET and its downstream signaling molecules, AKT, ERK, and S6 (Figure 3). Similarly, capmatinib showed clear inhibition of MET and its downstream signaling molecules in *MET*-amplified MKN45 and OE33 cells (Appendix A).

### 2.4. Effects of Savolitinib on MET-amplified GC MKN45 Xenografts

Next, we tested the efficacy of savolitinib in vivo. MKN45 xenografts were treated with savolitinib at the specified doses for 14 d (Figure 4a). Savolitinib monotherapy significantly inhibited the growth of xenografts (Figure 4b), but the body weights of mice were similar among the tested groups (Figure 4c). These findings suggest that savolitinib inhibits the growth of *MET*-amplified GC cells in vivo.

### 2.5. Combined Effects of Savolitinib and Trastuzumab on HER2/MET-Positive OE33 Cells

OE33 cells exhibit reduced sensitivity to the *HER2* inhibitor lapatinib when used as a single agent [24]. We previously confirmed their relative resistance to Tmab [25]. Tmab monotherapy does not significantly inhibit ERK and S6 activity on OE33 cells [25]. Consistently, *MET* co-amplifications are significantly associated with worse outcomes, indicating resistance to anti-HER2 targeted therapy for HER2+ GC in a post hoc analysis of 327 samples from the JACOB trial [26]. Here, treatment of OE33 cells with Tmab, savolitinib, or their combination effectively suppressed the growth of colonies (Figure 5a) and the downstream molecules, AKT, ERK, and S6 (Figure 5b). Therefore, a Tmab and savolitinib combination can effectively target *HER2/MET*-amplified OE33 cells.

## 3. Discussion

Our study demonstrated that the growth of *MET*-amplified GC cells was effectively suppressed by savolitinib and capmatinib. Here, we confirmed *MET* amplification in a large number of GC cell lines and these cell-based models could be useful for future studies of MET-targeted therapy for GC. In line with the earlier clinical trials, we showed that MET overexpression might not be the right biomarker for MET-targeted therapy in GC. *MET* inhibitors in combination with Tmab exhibited synergistic therapeutic effects on *HER2*- and *MET*-amplified GC cells. Although the incidence of *MET* amplification is relatively low in patients with GC, it could be a promising target for next-generation sequencing. 

Determining which patients with GC will benefit from c-MET inhibitors remains difficult. As discussed briefly, MET mAbs have failed in large phase III clinical trials, possibly because of the definition of MET-positive cases. Rilotumumab with epirubicin, cisplatin, and capecitabine (ECX) improved PFS to a median of 5.7 months compared to 4.2 months compared to ECX and the Hazard Ratio for PFS was 0.60 (80% confidence interval 0.45–0.79, *p* = 0.016) [27]. Subsequent phase III trial defined MET-overexpressing tumors when the tumor showed ≥25% of tumor cells with membrane staining of ≥1+ staining intensity [9]. *MET* gene amplification was assessed in exploratory analyses by fluorescence in situ hybridization (FISH) and amplification was defined as a *MET:CEN7* ratio of 2.0 or greater. The trial was terminated early because of the higher number of deaths in the rilotumumab group than in the placebo group. Moreover, no biomarkers (MET IHC, amplification, or *MET* copies) showed a distinct effect of rilotumumab. In the METGastric phase 3 trial, onartuzumab was evaluated as a first-line treatment in combination with mFOLFOX6 in patients with GC [10]. MET-positive tumors were defined by IHC as those where at least 50% of the tumor cells showed weak, moderate, and/or strong staining intensity (MET 1+/2+/3+). Unfortunately, this trial was terminated prematurely because of the lack of efficacy in a phase 2 trial [28]. The addition of onartuzumab to mFOLFOX6 did not improve OS, PFS, or ORR compared to mFOLFOX6 alone, irrespective of MET expression status. Based on these observations, we hypothesized that a positive IHC screening for cMET is not a suitable biomarker for patients with GC. The VIKTORY trial was a basket study of patients with GC based on clinical sequencing [29]. In this study, savolitinib, a class I c-MET inhibitor and small-molecule receptor TKI, showed an overall response rate of 50% (10 of 20) in the *MET*-amplified arm. This indicates that patients with *MET*-amplified GC can experience additional advantages from targeted therapy, potentially leading to improved OS compared to those who receive conventional second-line chemotherapy as part of their standard care. Although these *MET* TKIs show potential as targeted therapies for *MET*-amplified GC, none has been approved for GC treatment due to the lack of confirmatory clinical trials. Taken together, *MET* amplification could be the right biomarker for future clinical research of *MET* TKIs for GC.

*MET* amplification has been the subject of extensive research on both primary and acquired resistance to many types of targeted therapies. In total, 49 GC cell lines were screened for *MET* amplification, and for c-MET and p-MET [30]. We identified six GC cell lines with *MET* amplification and two c-MET-overexpressing cell lines. The authors also tested MET-targeting antibodies (Sym015 and SAIT301) and two small molecules (foretinib and tivantinib) in GC cells. However, none of these agents is currently approved for the treatment of GC. It is well known that *MET* amplification induces acquired resistance to gefitinib [31,32]. Hence, a treatment approach involving a MET inhibitor combined with an epidermal growth factor receptor (*EGFR*)-TKI could serve as a rational strategy to counter acquired MET-mediated resistance to *EGFR*-TKIs. In a recent TATTON trial, the combination therapy of savolitinib and osimertinib showed encouraging anti-tumor activity in patients with *MET*-amplified/overexpressed *EGFR*-mutated advanced NSCLC who had previously experienced disease progression on *EGFR* TKIs [33]. MET plays a role in Tmab resistance in *HER2*-positive breast cancer (BC) cells [34]. Inhibition of MET enhances the sensitivity of cells to Tmab-mediated growth inhibition, whereas MET activation protects cells against the effects of Tmab. These findings highlight the importance of MET in influencing the response to Tmab treatment in patients with HER2-positive BC. Similarly, HGF/MET-mediated acquired resistance to lapatinib has been reported as a mechanism of resistance to HER2-targeted agents in GC cells [35,36]. *MET* amplification has also been suggested as a candidate genomic alteration for predicting primary Tmab resistance in patients with HER2-positive GC [37]. In the present study, we thoroughly analyzed the protein expression and gene amplification of *MET* and *HER2* in 37 GC cell lines. Based on this analysis, we identified only *HER2/MET* dual-amplified OE33 cells, which are primarily resistant to Tmab [38], and suggested a potential combination of MET and HER2 inhibition to overcome Tmab resistance. A previous study of *MET*-amplified GC cells did not include a dual-positive cell line to test this combination [30].

Even though there are no approved therapies for *MET*-amplified GC, a few studies have addressed potential resistance mechanisms to *MET* TKIs. Following the VIKTORY trial, they identified that three patients with *MET*-amplified GC initially responded to savolitinib but later developed acquired resistance [39]. Using a next-generation sequencing 100-gene panel, they identified key resistance mechanisms including *MET* D1228V/N/H and Y1230C mutations, as well as high copy number *MET* gene amplifications. Other researchers reported the overexpression of FGFR2 as an intrinsic resistance to MET inhibitors with patient-derived gastric cancer xenograft models [40]. This study utilized PHA665752, a small molecule *MET* TKI; however, clinical development involving this compound is currently inactive. Further research is warranted to explore FGFR2 overexpression as a potential resistance mechanism to approved *MET* TKIs. 

MET inhibitors are effective against MET-driven cancers; however, they also pose toxicity concerns. Capmatinib, crizotinib, and tepotinib monotherapies are approved for the treatment of metastatic NSCLC with high-level *MET* exon 14 skipping mutations. In the GEOMETRY mono-1 trial with capmatinib [19], 66% of the patients experienced emergent adverse events (TEAEs) of grade ≥3. Serious adverse reactions were observed in 51% of patients, with dyspnea (7%), pneumonia (4.8%), and pleural effusion (3.6%) being the most commonly reported. Tragically, one patient (0.3%) experienced a fatal adverse reaction attributed to pneumonitis. In the VISION trial with tepotinib [41], 34.8% of the patients encountered TEAEs of grade ≥3. Peripheral edema was the most prevalent TEAE (67.1%), with 11.2% of the patients experiencing grade ≥3 peripheral edema. Fatal adverse reactions, including pneumonitis, hepatic failure, and dyspnea from fluid overload, were observed in one patient (1.0%). Amivantamab-vmjw (amivantamab) is a bispecific EGFR/MET antibody approved for patients with advanced NSCLC with *EGFR* exon 20 insertion mutations after therapy [42]. Serious adverse reactions were noted in 30% of the patients. Permanent discontinuation due to adverse reactions occurred in 11% of patients, with the most frequent (≥1%) adverse reactions leading to permanent discontinuation being pneumonia, infusion-related reaction, pneumonitis/interstitial lung disease, dyspnea, pleural effusion, and rash. Fatal adverse reactions were reported in three patients (2.3%), with two fatal events attributed to pneumonia and one recorded as sudden death. The safety profiles revealed by these findings are unsatisfactory for targeted agents. However, these findings provide important insights for the careful selection of patients for MET-targeted therapy and facilitate the development of better inhibitors with low toxicity.

Our study has limitations. First, we were not able to test all of the *MET* TKIs on the horizon. For example, tepotinib was recently approved for patients with NSCLC harboring *MET* exon 14 skipping alterations by the US FDA. This agent could also be effective for *MET*-amplified GC cells. Secondly, the combination of HER2 and MET inhibitors for dual-positive GC cells was not tested in a xenograft mouse model. Despite promising efficacy shown in cell culture-based assays, several studies have failed to confirm the synergistic effect in xenograft experiments. Furthermore, the present study cannot suggest the best approach for patients with *MET*-amplified GC. Capmatinib, crizotinib, and tepotinib have received approval from the US FDA for NSCLC with *MET* alterations, while savolitinib is approved in China. As *MET* alterations in NSCLC patients are relatively rare, there are no studies available for comparing these agents. Although our data demonstrated the promising efficacy of *MET* TKIs for *MET*-amplified GC cells, approval by regulatory authorities will necessitate results from clinical trials. Given the rarity of *MET* amplification in GC, fostering global collaboration will enhance the development of these agents in clinical trials.

## 4. Materials and Methods

### 4.1. Cell Lines and Reagents

Among the 37 human gastric and esophageal adenocarcinoma cell lines obtained for this study [25], three GC (SNU-620, MKN-45, and SNU-5) and two esophageal adenocarcinoma (ESO51 and OE33) cell lines were selected for analysis. AGS, MKN-45, SNU-620, SNU638, and SNU-5 cells were obtained from the Korean Cell Line Bank (Seoul, Republic of Korea). ESO51 and OE33 cells were purchased from Sigma Aldrich (St. Louis, MO, USA). YCC-19, YCC-33, and YCC-38 cell lines were provided by Professor Sun-Young Rha (Yonsei Cancer Center, Republic of Korea). NUGC-4 cells were purchased from the RIKEN BRC Cell Bank (Tsukuba, Ibaraki, Japan). All cells were cultured in the Roswell Park Memorial Institute-1640 (RPMI-1640) medium containing 10% fetal bovine serum, 100 U/mL penicillin, and 100 mg/mL streptomycin. Cultured cells were maintained at 37 °C in an atmosphere of 5% CO_2_. Trastuzumab was purchased from Orient Bio (Gyeonggi-do, Republic of Korea). Capmatinib (#S2788) and savolitinib (#S7674) were purchased from Selleck Chemicals (Houston, TX, USA).

### 4.2. ddPCR

Next, ddPCR assay was performed by the LOGONE Bio-Convergence Research Foundation (Seoul, Republic of Korea) using the QX200 system (Bio-Rad, Hercules, CA, USA), according to the manufacturers’ recommendations. The reaction mixture in a volume of 20 µL consisted of 4× ddPCR multiplex supermix (5 μL) for Probes (Bio-Rad, Hercules, CA, USA), primers, probes, and DNA template. For MET copy number analysis, the concentrations of primers and probes were kept in the range of 900–250 nM. All primer sequences are listed in Appendix A. All procedures were as previously described [25]. CNV values were determined from the copy number of each reference gene using the following equation:(1)$$Copy\;number\;variation=\frac{MET\;copies}{Average\;copies\;of\;five\;reference\;genes}\times 2$$

### 4.3. Cell Proliferation Assay

Cells (5 × 10^3^ cells/well) were seeded in 96-well plates and treated with capmatinib or savolitinib for a specific period. Cell survival rate was measured using the cell counting kit-8 (CCK-8; Dojindo, Japan) and calculated using the following formula: ([A450 of experimental group − A450 of background group]/[A450 of control group − A450 of background group]) × 100. 

### 4.4. Clonogenic Cell Survival Assay and Culture in Soft Agar

Clonogenic assays were performed as previously described [43]. Cells at a density of 2500–5000 cells per well were seeded overnight in 24-well plates. The next day, the cells were treated with capmatinib or savolitinib. The medium was replaced with a fresh medium every three days. Control cells were treated with an equivalent amount of dimethyl sulfoxide. After 10–14 d, the cells were stained with a crystal violet solution (Cat. No. V5265; Sigma-Aldrich, St Louis, MI, USA) diluted with distilled water to a concentration of 0.1%. The cells were destained with distilled water and quantified using the ImageJ Software 1.53i (National Institutes of Health, Bethesda, MA, USA). For the anchorage-independent colony formation assay, 2.5 × 10^3^ to 7.5 × 10^3^ cells were suspended in 0.5 mL of 0.4% top agar that was layered on top of 1 mL of 1% base agar in each well in 12-well multi-well plates, as previously described [25]. 

### 4.5. Western Blotting Analysis

Cells (2–8 × 10^5^) were seeded in 100 mm dishes. Proteins were harvested as previously described. Western blotting was performed to determine the expression of relevant signaling pathways, such as the HER2 signaling pathway. Primary antibodies against the following molecules were purchased from Cell Signaling Technology (Beverley, MA, USA): p-MET-Tyr 1234/1235 (#3126), MET (#4560), p-HER2-Tyr 1221/1222 (#2249), HER2 (#2165), p-AKT-Ser 473 (#4060), AKT (#2217), p-ERK-Thr 202/Tyr 204 (#9106), p-S6-Ser 235/236 (#2211), and S6 (#2217). Antibodies against AMPK (#sc-19128), ERK (#sc-271269), and actin (#sc-47778) were purchased from Santa Cruz Biotechnology (Dallas, TX, USA). Secondary antibodies were purchased from Thermo Fisher Scientific (Waltham, MA, USA). Membranes were blocked with the blocking buffer (5% non-fat dry milk in Tris-buffered saline containing 0.01% Tween-20 (TBST]) for 1 h at room temperature and incubated with primary antibodies diluted in 5% bovine serum albumin in TBST (1:1000) overnight at 4 °C. The membranes were washed multiple times with TBST and incubated again with the corresponding horseradish peroxidase-conjugated secondary antibodies diluted in 3% non-fat dry milk in TBST (1:5000) for 1 h at room temperature. The membranes were washed multiple times with PBST and visualized using an enhanced chemiluminescence detection kit (Amersham Biosciences, Arlington Heights, IL, USA).

### 4.6. Establishment of a Xenograft Mouse Model

All animal procedures were approved by and performed according to the protocols of the Seoul National University Institutional Animal Care and Use Committee (SNU-200622-1). Five-week-old athymic nude mice were obtained from Orient Bio (Seongnam-si, Republic of Korea) and acclimated for 14 d. Human MKN-45 GC cells were cultured in the RPMI-1640 medium containing 10% fetal bovine serum and resuspended in 50% Cultrex Basement Membrane Extract in phosphate-buffered saline. The mice were subcutaneously injected with viable MKN-45 cells (2 × 10^6^ cells/mouse) into the right flank. When the average tumor size reached approximately 150 mm^3^, the mice were randomly divided into vehicle control and treatment groups, with six mice per group. The day of randomization was defined as day 0. The size of the tumor was measured twice weekly with digital calipers, and the volume of the tumor was calculated using the following formula: volume (mm^3^) = length (mm) × width (mm)^2^ × 0.5. All mice were observed daily during the treatment period.

### 4.7. Statistical Analyses

Statistical analyses were conducted using SPSS version 29.0 (IBM SPSS Statistics for Windows; IBM Corp., Armonk, NY, USA). Data are presented as means ± standard error (SE). The statistical significance of differences was determined using the Kruskal–Wallis test, one-way analysis of variance, and Student’s *t*-test, as appropriate. All statistical analyses were two-tailed. Statistical significance was set *p* < 0.05.

## 5. Conclusions

In conclusion, our findings demonstrated that the *MET* TKIs savolitinib and capmatinib effectively suppressed the proliferation of *MET*-amplified human GC cells and inhibited tumor growth. GC cells with MET overexpression showed a variable response to *MET* TKIs, while those with *MET* amplification are consistently sensitive. Based on this result, we recommend using *MET* amplification as a biomarker in clinical research involving *MET* TKIs for GC. Furthermore, further investigation is necessary to explore the potential benefits of combining anti-HER2 agents with *MET*-TKIs for *HER2*- and *MET*-amplified GC treatment. Although currently available MET-targeted agents show therapeutic potential for patients with specific types of MET-driven cancers, further research is necessary to develop strategies to overcome the significant toxicity of MET inhibitors. 

## Figures and Tables

**Figure 1 ijms-25-05975-f001:**
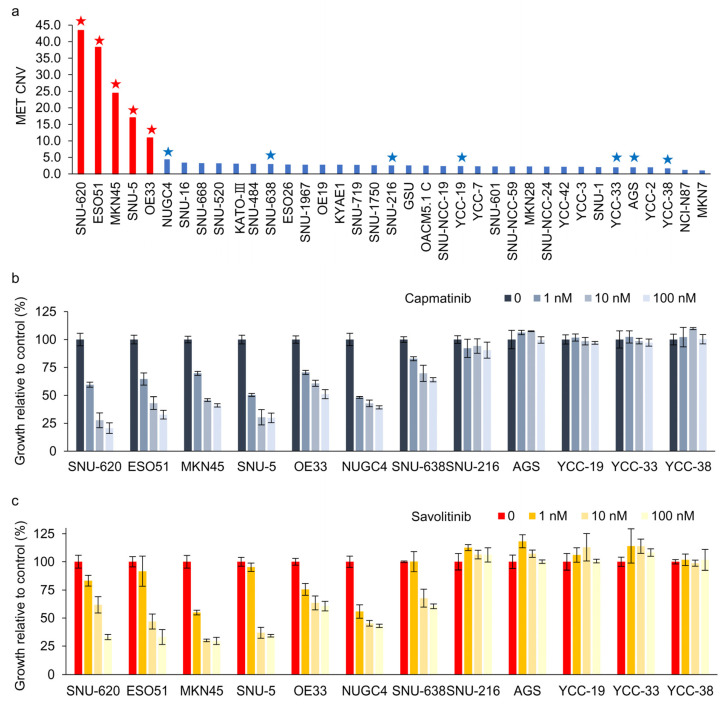
Dose-dependent effects of capmatinib and savolitinib on *MET*-positive and -negative gastric cancer (GC) cells. (**a**) Droplet digital polymerase chain reaction (ddPCR) was performed to determine the MET copy number variations in 37 GC cell lines. *MET*-amplified cell lines are indicated by red bars. Cells marked with stars were selected for subsequent cell counting kit (CCK)-8 assays. (**b**,**c**) GC cell lines were treated with the specified concentrations of capmatinib and savolitinib, and CCK-8 assays were performed after 72 h (*n* = 3).

**Figure 2 ijms-25-05975-f002:**
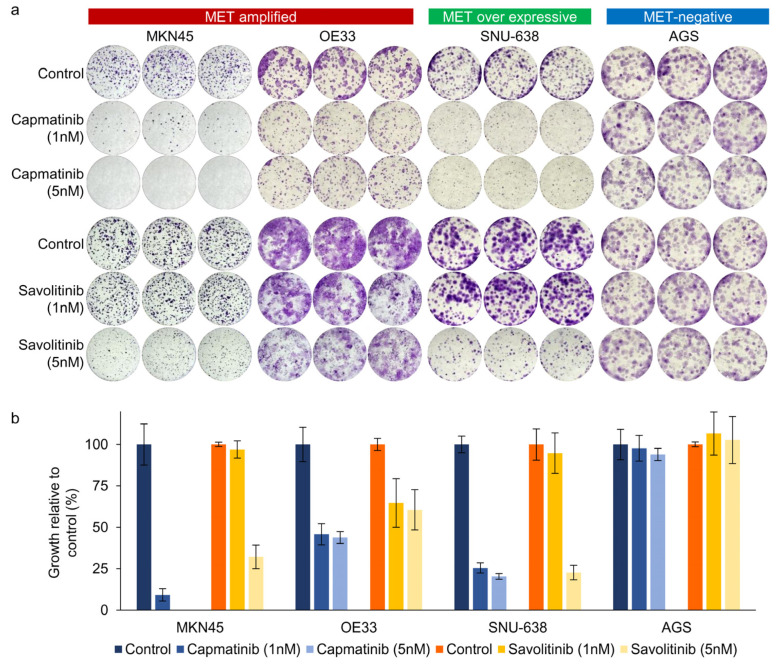
Long-term effects of capmatinib and savolitinib on MET-amplified, MET-overexpressing, and MET-negative GC cells determined via clonogenic cell survival assays (**a**). Relative colony counts of GC cells treated with the indicated concentrations of capmatinib and savolitinib for 2 weeks (**b**).

**Figure 3 ijms-25-05975-f003:**
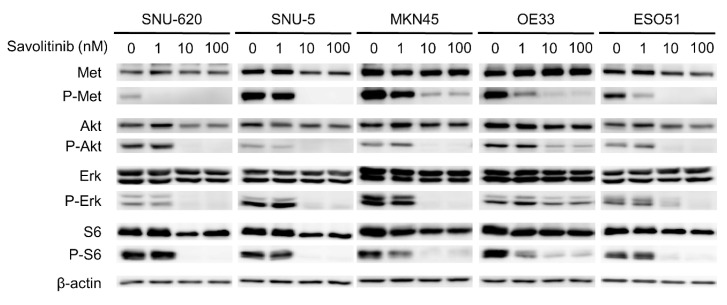
Effects of savolitinib on *MET*-amplfied GC cells assessed via immunoblotting analysis. Immunoblotting analysis using antibodies specific for the proteins cell lysates treated with savolitinib (1, 10, and 100 nM) for 24 h before harvest. A representative experiment showed that savolitinib effectively inhibited the phosphorylation of MET and its downstream pathways.

**Figure 4 ijms-25-05975-f004:**
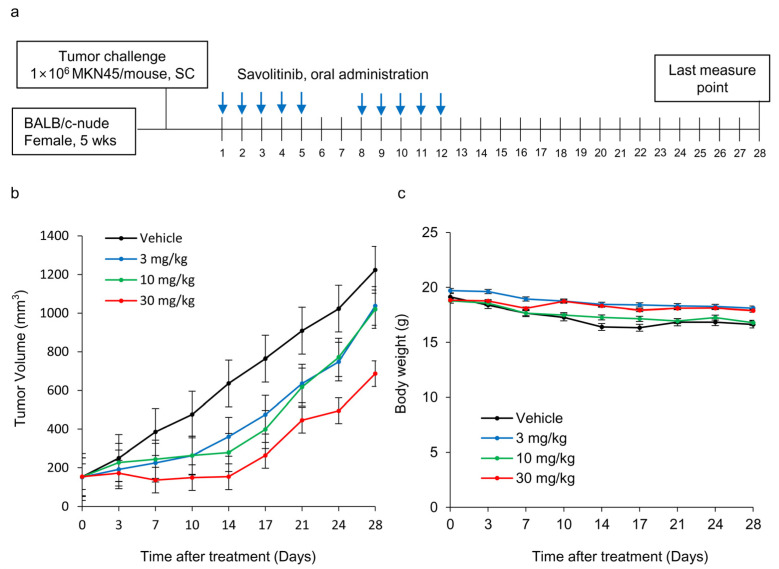
Effects of savolitinib on *MET*-amplified MKN45 xenografts. (**a**) Dosing schedule for savolitinib. Mice were administered savolitinib via oral administration for 2 weeks. (**b**) During the drug administration period (14 d), tumor growth was significantly inhibited in the treated cells compared to that in the vehicle. However, tumor recurrence was observed after discontinuation of treatment. (**c**) Body weights of mice were similar in all groups.

**Figure 5 ijms-25-05975-f005:**
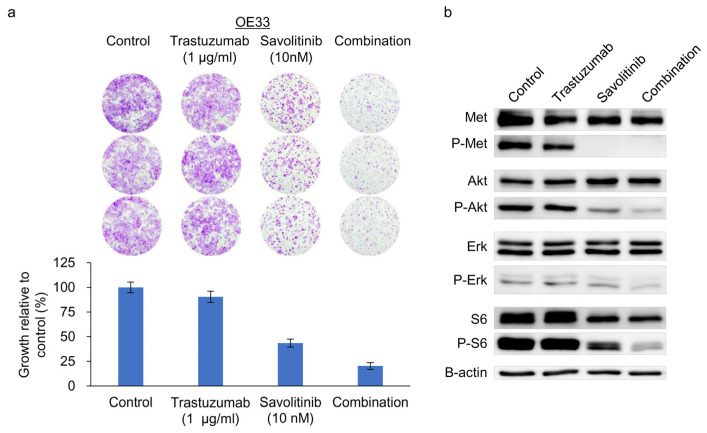
Clonogenic cell survival and immunoblot assays of *HER2*- and *MET*-amplified OE33 cells. (**a**) Cells were treated with trastuzumab and capmatinib, alone and in combination, for 14 d, and clonogenic assays were performed. (**b**) Immunoblotting analysis using antibodies specific for the proteins of lysates treated with trastuzumab and savolitinib alone or in combination for 24 h before harvest. Combined treatment with savolitinib inhibited the activation of extracellular signal-regulated kinase (ERK)-1/2 and protein kinase B (AKT) via MET dephosphorylation in OE33 cells.

## Data Availability

The data presented in this study are available upon request from the corresponding author.

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
