# Peer review of "Characterization of MET Alterations in 37 Gastroesophageal Cancer Cell Lines for MET-Targeted Therapy"

_ijms, 2024, doi:10.3390/ijms25115975_

Round 1
Reviewer 1 Report
Comments and Suggestions for Authors
· I believe the introduction is too brief and should be strengthened with more references. There are missing references to literature, which are closely related to this work. Please refer to these articles (J Med Chem. 2022 Nov 24;65(22):15140-15164; JCO Precis Oncol. 2020 Mar 24:4:PO.19.00386; Invest New Drugs. 2014 Apr;32(2):355-61; BMC Cancer. 2016 Jan 22:16:35; Cancers (Basel). 2020 Jun 15;12(6):1575; Mol Cancer Ther. 2015 Feb;14(2):384-94; Oncol Lett. 2015 Oct;10(4):2003-2008; Invest New Drugs. 2020 Dec;38(6):1633-1640).
· The novelty of the study is very weak. I would suggest that the authors to present the aim of the paper with regards to what is currently known by in vivo/vitro and RCTs studies, therefore highlighting the added value of this study. This should be clearly addressed in the last paragraph of introduction.
· It is unclear if this study evaluate the efficacy of capmatinib and savolitinib against MET-75 amplified GC cell lines in vitro, in vivo, or both (Line 75-76).
· The sentence spanning from 68 to 74 should be moved to discussion or conclusion.
· Why cell apoptosis and viability assay were not evaluated in this study? The activation of the Akt and ERK signaling pathways antagonize capmatinib and savolitinib-induced gastric cancer cell apoptosis.
· Why the toxicity profile of c-MET inhibitors in GC cells is not assessed?
· The discussion provided insufficient or too speculative interpretation of the results. Without connection to the literature (refer to my comment in introduction with regards to missing studies on the topic), the meaning of this study findings cannot be interpreted validly.
· It does not appear that the authors noted any limitations of the study. This should be in a separate section.
· Please include a list of abbreviations at the end.
Comments on the Quality of English LanguageLanguage needs to be improved throughout. Please consult with an English language speaker or MDPI Author service.
Author Response
Reviewer 1
- I believe the introduction is too brief and should be strengthened with more references. There are missing references to literature, which are closely related to this work. Please refer to these articles (J Med Chem. 2022 Nov 24;65(22):15140-15164; JCO Precis Oncol. 2020 Mar 24:4:PO.19.00386; Invest New Drugs. 2014 Apr;32(2):355-61; BMC Cancer. 2016 Jan 22:16:35; Cancers (Basel). 2020 Jun 15;12(6):1575; Mol Cancer Ther. 2015 Feb;14(2):384-94; Oncol Lett. 2015 Oct;10(4):2003-2008; Invest New Drugs. 2020 Dec;38(6):1633-1640).
Answer>
As the reviewer suggested, we added a paragraph to cover various MET inhibitors tested for gastric cancers. Two references about resistance to MET inhibitors are referred in the Discussion. All the references are now included in the revision.
- The novelty of the study is very weak. I would suggest that the authors to present the aim of the paper with regards to what is currently known by in vivo/vitro and RCTs studies, therefore highlighting the added value of this study. This should be clearly addressed in the last paragraph of introduction.
Answer>
As the reviewer suggested, we added several sentences to clarify the aim of this study in the last paragraph in the Introduction.
- It is unclear if this study evaluate the efficacy of capmatinib and savolitinib against MET-75 amplified GC cell lines in vitro, in vivo, or both (Line 75-76).
Answer>
We tested both capmatinib and savolitinib in vitro with the MET amplified GC cells. For xenografts experiment, we only tested savolitinib. We revised the description for the clarification.
- The sentence spanning from 68 to 74 should be moved to discussion or conclusion.
Answer>
As the reviewer suggested, the sentences were moved to discussion.
- Why cell apoptosis and viability assay were not evaluated in this study? The activation of the Akt and ERK signaling pathways antagonize capmatinib and savolitinib-induced gastric cancer cell apoptosis.
Answer>
We tested the growth inhibition of the included GC cells by CCK-8 assays and clonogenic cell survival assays. As you may understand, CCK-8 assays are known to be one of the viability assays. But, CCK-8 assays only test the cells for 3 days. We tried to confirm the long-term effect of the agents by clonogenic cell survival assays. We did not perform Western blot for caspase cleavage or PARP, but these two independent viability assays showed similar results.
Since we are evaluating the efficacy of MET signaling inhibitors, we showed the downstream pathway activity by Western blotting. The downstream includes AKT, ERK and S6, and the phosphorylation of these protein is well correlated with the activity.
- Why the toxicity profile of c-MET inhibitors in GC cells is not assessed?
Answer>
The toxicity profile of investigational new drugs cannot be tested with cancer cell line-based experiments, since the toxicities are mostly from the normal organs. Most toxicity screening for the developmental therapeutics are performed on animal experiment.
- The discussion provided insufficient or too speculative interpretation of the results. Without connection to the literature (refer to my comment in introduction with regards to missing studies on the topic), the meaning of this study findings cannot be interpreted validly.
Answer>
As the reviewer suggested, we added a paragraph to cover various MET inhibitors tested for gastric cancers. All the references are now included in the revision. We emphasized findings from our study.
- It does not appear that the authors noted any limitations of the study. This should be in a separate section.
Answer>
As the reviewer suggested, we added a paragraph to describe the limitation of this study.
- Please include a list of abbreviations at the end.
Answer>
As the reviewer suggested, we added a list of abbreviations at the end.
Reviewer 2 Report
Comments and Suggestions for Authors
Jin-Soo et al. submitted the manuscript entitled: Characterization of MET alterations in 37 gastroesophageal cancer cell lines for MET-targeted therapy, in which they identified several cell lines with abnormally high MET copy. Afterwards, the authors applied two clinically approved MET inhibitor Capmatinib and Savolitinib to these cell lines and monitored cell growth, colony formation, changes in protein phosphorylation level and in vivo bioevaluation. The authors also proved syngenetic effect of HER2 mAb and METi. Generally, this is a well-prepared manuscript and this topic will be of interest to potential readers of IJMS.
I only have some minor comments.
1. Page 4, figure 2: In section 2.1, my understanding is the authors tried to distinguish between MET overexpression and amplification when METi was applied in the GC treatments. But from the data of capmatinib and savolitnib, MET overexpression can also response METi treatment. It is worth discussing more about this phenomenon.
2. Discussion: Undoubtedly the authors provided a comprehensive analysis of potential clinical significance of METi. But in the same time the author also should provide a general summary of the work in this manuscript. Current version of manuscript only mentioned the synergic effect of Her2 mAb and METi, but other findings like MET copy numbers correlate with METi response is also worth mentioning in discussion section.
3. A typo exists at abstract, line 13.
Author Response
Jin-Soo et al. submitted the manuscript entitled: Characterization of MET alterations in 37 gastroesophageal cancer cell lines for MET-targeted therapy, in which they identified several cell lines with abnormally high MET copy. Afterwards, the authors applied two clinically approved MET inhibitor Capmatinib and Savolitinib to these cell lines and monitored cell growth, colony formation, changes in protein phosphorylation level and in vivo bioevaluation. The authors also proved syngenetic effect of HER2 mAb and METi. Generally, this is a well-prepared manuscript and this topic will be of interest to potential readers of IJMS.
Answer>
Thank you very much for the kind and encouraging evaluation as well as critical comments. We did our best to address your comments.
I only have some minor comments.
- Page 4, figure 2: In section 2.1, my understanding is the authors tried to distinguish between MET overexpression and amplification when METi was applied in the GC treatments. But from the data of capmatinib and savolitnib, MET overexpression can also response METi treatment. It is worth discussing more about this phenomenon.
Answer>
As the reviewer pointed out, GC cells with MET overexpression also showed modest response to MET TKIs. We described this results in the section 2.2, “Overall, MET-overexpressing GC cells showed variable sensitivity, whereas MET-positive GC cells showed high sensitivity to MET TKIs.” The discrepancy is the reason why MET targeting antibodies failed in clinical trials. Also, we emphasize our finding of MET amplification over overexpression in the first paragraph of discussion.
- Discussion: Undoubtedly the authors provided a comprehensive analysis of potential clinical significance of METi. But in the same time the author also should provide a general summary of the work in this manuscript. Current version of manuscript only mentioned the synergic effect of Her2 mAb and METi, but other findings like MET copy numbers correlate with METi response is also worth mentioning in discussion section.
Answer>
As the reviewer suggested, we added sentences in the first paragraph to discuss more about the results from our manuscript.
- A typo exists at abstract, line 13.
Answer>
As the reviewer suggested, we corrected typo in the abstract. We also thoroughly reviewed the manuscript again for the typo error.
Reviewer 3 Report
Comments and Suggestions for Authors
Reference Report
Title: Characterization of MET alterations in 37 gastroesophageal cancer cell lines for MET-targeted therapy
Manuscript number: ijms-3037026
Submitted to IJMS
By Kim et al
This study investigated MET alterations in 37 gastroesophageal cancer cell lines for targeted therapy. I have the following concerns regarding this work:
- Introduction: The introduction provides a good and comprehensive background and motivation for the study. However, the authors should review and include some related works on MET characterization in cancer cells for targeted therapy.
- Figure 1: Ensure consistency in labeling by using either "1A" or "1a" throughout the text. Apply the same approach to the remaining figures.
- Figure 4: Please include error bars for Figure 4.
- Section 4.2, Line 26: Please label the equation provided in this section.
- Materials and Methods: The authors need to thoroughly explain how they selected the 37 cancer cell lines used in this study.
- Future Work and Limitations: The authors should discuss any potential future work following their findings and mention the limitations of their study.
It needs careful edition of the English.
Author Response
Reviewer 3
Introduction: The introduction provides a good and comprehensive background and motivation for the study. However, the authors should review and include some related works on MET characterization in cancer cells for targeted therapy.
Answer>
As the reviewer suggested, we added a paragraph to cover various MET inhibitors tested for gastric cancers.
Figure 1: Ensure consistency in labeling by using either "1A" or "1a" throughout the text. Apply the same approach to the remaining figures.
Answer>
As the reviewer suggested, we corrected the labeling.
Figure 4: Please include error bars for Figure 4.
Answer>
As the reviewer suggested, we added the error bars.
Section 4.2, Line 26: Please label the equation provided in this section.
Answer>
As the reviewer suggested, we added the label for the equation.
Materials and Methods: The authors need to thoroughly explain how they selected the 37 cancer cell lines used in this study.
Answer>
The cell lines were acquired over several years upon the availability. As mentioned in the Materials and Methods, most cell lines were obtained from the Korean Cell Line Bank (KCLB, Seoul, Republic of Korea) or given by Prof. Sun-Young Rha from Yonsei University, Korea. For the previous publication (Kim JS, Kim MY, Hong S. Synergistic Effects of Metformin and Trastuzumab on HER2 Positive Gastroesophageal Adenocarcinoma Cells In Vitro and In Vivo. Cancers (Basel). 2023;15), we tried to obtain HER2 positive GC cells worldwide.
Future Work and Limitations: The authors should discuss any potential future work following their findings and mention the limitations of their study.
Answer>
As the reviewer suggested, we added a paragraph to describe the limitation of this study.
Round 2
Reviewer 1 Report
Comments and Suggestions for Authors
No further comments.
Comments on the Quality of English LanguageLanguage editing required throughout.
Reviewer 3 Report
Comments and Suggestions for Authors
I am satisfied with the additional contents and modifications as per my comments. The quality and presentation of this manuscript are improved.
Comments on the Quality of English LanguageNo comment.